# Perturbation-response genes reveal signaling footprints in cancer gene expression

Michael Schubert[1], Bertram Klinger[2,3], Martina Klünemann[2,3], Anja Sieber[2,3], Florian Uhlitz [2,3], Sascha Sauer[4], Mathew J. Garnett[5], Nils Blüthgen [2,3] & Julio Saez-Rodriguez [1,6]

Aberrant cell signaling can cause cancer and other diseases and is a focal point of drug research. A common approach is to infer signaling activity of pathways from gene expression. However, mapping gene expression to pathway components disregards the effect of post-translational modifications, and downstream signatures represent very specific experimental conditions. Here we present PROGENy, a method that overcomes both limitations by leveraging a large compendium of publicly available perturbation experiments to yield a common core of Pathway RespOnsive GENes. Unlike pathway mapping methods, PROGENy can (i) recover the effect of known driver mutations, (ii) provide or improve strong markers for drug indications, and (iii) distinguish between oncogenic and tumor suppressor pathways for patient survival. Collectively, these results show that PROGENy accurately infers pathway activity from gene expression in a wide range of conditions.

[1] European Molecular Biology Laboratory, European Bioinformatics Institute, Wellcome Genome Campus, Cambridge, CB10 1SD, UK. [2] Institute of Pathology, Charité Universitätsmedizin Berlin, Charitéplatz 1, 10117 Berlin, Germany. [3] IRI Life Sciences and Institute for Theoretical Biology, Humboldt University Berlin, Philippstr. 13/Haus 18, 10115 Berlin, Germany. [4] Max Delbrück Center for Molecular Medicine (MDC), Berlin Institute for Medical Systems Biology/Berlin Institute of Health, Robert-Rössle-Str. 10, 13092 Berlin, Germany. [5] Wellcome Trust Sanger Institute, Wellcome Genome Campus, Cambridge CB10 1SA, UK. [6] RWTH Aachen University, Faculty of Medicine, Joint Research Centre for Computational Biomedicine, Aachen 52057, Germany. Correspondence and requests for materials should be addressed to J.S-R. (email: saezrodriguez@gmail.com)

A wealth of molecular data have become available that reflects a cell's state in different diseases. The challenge that remains is how to derive predictive and reliable biomarkers for disease status, treatment opportunities, or patient outcome in a way that is both relevant and interpretable. Of particular interest are methods that infer and quantify deregulation of signaling pathways, as those are key for many processes underpinning different diseases.

A particular example of this is cancer, which is largely caused by cell signaling aberrations created by driver mutations and copy number alterations[1]. Here, efforts like the TCGA[2] and ICGC[3] have pioneered molecular characterization of primary tumors on a large scale. In addition, the GDSC[4,5] and CCLE[6] have focussed on preclinical biomarkers of drug sensitivity in cancer cell lines. These initiatives have provided profound insight in the molecular markup of the disease. However, putting the genomic alterations investigated in the functional context of the pathways they alter may provide additional information on mechanisms of pathogenesis and treatment opportunities[7].

With direct measurements of signaling activity not widely available, the latter has often been inferred using gene expression. This includes quantifying the expression level of a pathway gene set (e.g., Gene Ontology[8] or Reactome[9]) using Gene Set Enrichment Analysis[10], or other methods that are able to take pathway structure into account[11–13]. While these methods can be applied to almost any pathway, they are based on mapping transcript expression to the corresponding signaling proteins and hence do not take into account the effect of post-translational modifications (Fig. 1a). It is therefore unclear if and under what circumstances the pathway scores obtained by these methods reflect signaling activity.

A complementary approach is to contrast two conditions with known differential activity by means of a gene expression signature[14]. Of particular interest are short-term perturbation experiments that capture the primary response to a stimulus. A well-known example of this is the Connectivity Map[15] that has been used to match drug-induced gene expression changes for disease indications or drug repurposing[16]. In a similar manner, many signatures have been proposed to infer pathway activity[17–23], including seminal work by Bild et al.[17] that was later also used to predict drug response in breast cancer cell lines[18,24].

However, the same signaling pathways may trigger different downstream gene expression programs depending on the cell type or the perturbing agent used. Hence, if gene expression signatures are to be used as a generally applicable pathway method, there is a need to address this context specificity. In the past, methods have been developed that addressed this by building consensus models over multiple signatures and using these to infer pathway activity[5,25,26]. These methods, however, have been limited by a low number of perturbation experiments as well as inherent application constraints.

Here, we overcome the limitations of both approaches by leveraging a large compendium of publicly available perturbation experiments that yield a common core of Pathway RespOnsive GENes to a specified set of stimuli. PROGENy is able to better infer pathway activity from perturbation experiments than EPSA[25], is applicable to panels of samples unlike SPEED[26], and performs better than a previous extension we proposed to the latter[5].

We performed a systematic comparison of PROGENy and other commonly used pathway methods for 11 cancer-relevant pathways. We investigated how well each method can recover pathway perturbations and constitutive activity mediated by driver mutations in The Cancer Genome Atlas (TCGA)[2]. We further examined how well they can explain drug sensitivity to 265 drugs in 805 cancer cell lines in the Genomics of Drug Sensitivity in Cancer (GDSC)[4,5] and patient survival in 7254 primary tumors spanning 34 tumor types using TCGA data. We found that PROGENy significantly outperforms existing methods for these tasks.

## Results

**Consensus gene signatures for pathway activity.** We curated (workflow in Fig. 1b; experiments in Supplementary Note 1) a total of 208 different submissions to ArrayExpress/GEO, spanning perturbations of the 11 pathways EGFR, MAPK, PI3K, VEGF, JAK-STAT, TGFb, TNFa, NFkB, Hypoxia, p53-mediated DNA damage response, and Trail (apoptosis). Our data set consists of 568 experiments and 2652 microarrays, making it the largest study of pathway signatures to date (Fig. 1c and Supplementary Fig. 1).

We calculated $z$-scores of gene expression changes for each experiment, for which we trained a regression model using the perturbed pathway as input and gene expression as a response variable. For each pathway, we identified 100 responsive genes that are most consistently deregulated across experiments (Supplementary Fig. 2). These responsive genes are specific to the perturbed pathway and have little overlap with genes encoding for its signaling proteins (Supplementary Fig. 3). This underscores the fact that pathway expression and activation are distinct processes and suggests that they should be treated separately. We use the $z$-scores of those 100 pathway-responsive genes in a simple, yet effective, linear model to infer pathway activity from gene expression called PROGENy (for Pathway RespOnsive GENes, but also to indicate the descent of the method from previously published experiments; Supplementary Data 1). We find that our responsive genes are often enriched in biological processes related to a signaling pathway, but not the pathway itself (Supplementary Fig. 4).

Using a leave-one-out strategy of model building and perturbation scoring, our inferred pathway activation is strongly ($p < 10^{-10}$, except $p < 10^{-5}$ for Trail) associated with the pathway that was experimentally perturbed. The associations of a pathway signature with other pathways are weaker ($p > 10^{-5}$), except for EGFR with MAPK/PI3K and TNFa with NFkB/MAPK (Fig. 2a and Supplementary Fig. 5, left), where there is biologically known cross-activation[27]. Relative activation patterns are consistent across input experiments (Supplementary Fig. 5, right).

PROGENy separates basal and perturbed arrays better (Supplementary Table 1; binomial test; $p < 0.04$) than EPSA[25] on our curated set of experiments, and in addition to SPEED[26] also infers the sign of pathway activity (Supplementary Fig. 6). We find that building the consensus of many experiments is essential, as the $z$-scores from a single experiment perform no better than random, and using too few experiments to derive the model degrades performance. The exact number of experiments required differs between pathways, but we see a plateau effect between 20 and 50 signatures for most of them.

In order to also test PROGENy on a completely separate set, we set aside 10 perturbation experiments that also measured pathway activity in an orthogonal manner. We compared the activity measurement from basal and perturbed condition with the pathway scores that PROGENy inferred, and found that our method could always predict the direction of perturbation correctly, with separation statistics that are comparable to direct measurements (Supplementary Fig. 7). Furthermore, we performed independent validation experiments using the HEK293-ER cell line, where we performed 5 distinct pathway perturbations. We induced RAF/MAPK signaling using 4-hydroxy tamoxifen (4OHT) that stimulates an RAF-ER transgene, and used the PI3K inhibitor Ly294002 to block the PI3K/AKT

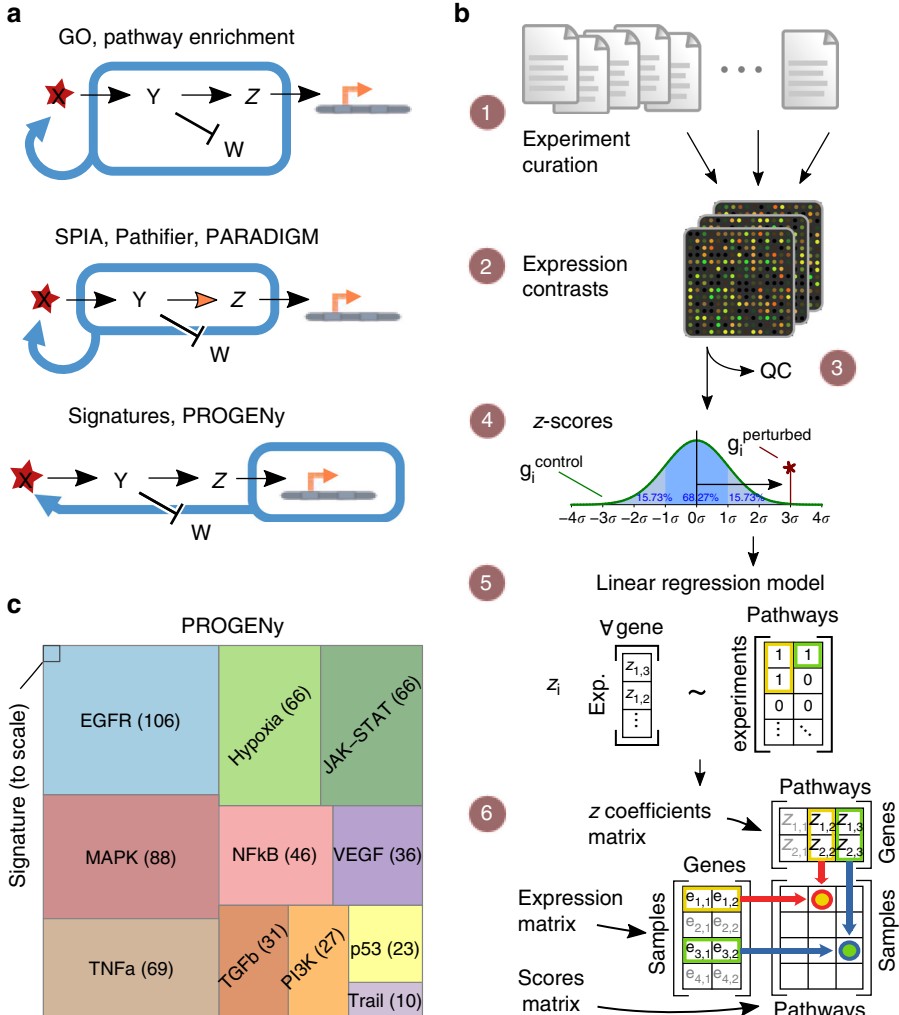

**Fig. 1** Deriving pathway-response signatures for 11 pathways. **a** Reasoning about pathway activation. Most pathway approaches make use of either the set (top panel) or infer or incorporate structure (middle panel) of signaling molecules to make statements about a possible activation, while signature-based approaches such as PROGENy consider the genes affected by perturbing the pathway. **b** Workflow of the data curation and model building. (1) Finding and curation of 208 publicly available experiment series in the ArrayExpress database, (2) Extracting 556 perturbation experiments from series' raw data, (3) Performing QC metrics and discarding failures, (4) Computing z-scores per experiment, (5) Using a linear regression model to fit genes responsive to all pathways simultaneously obtaining the z-coefficients matrix, (6) Assigning pathway scores using the coefficients matrix and basal expression data. See methods section for details. **c** Size of the data set compared to an individual gene expression signature experiment. The amount of experiments that comprise each pathway is shown to scale and indicated. Figure 1b (2) created by Guillaime Paumier is published under a CC-BY-SA license, sourced from https://commons.wikimedia.org/wiki/File:DNA_microarray.svg. Figure 1b (4) is an adaptation (by Chen-Pan Liao) of the original work of User:Jhguch at en.wikipedia, published under a CC-BY-SA license, sourced from https://commons.wikimedia.org/wiki/File:Boxplot_vs_PDF.svg. Figure 1b (6) is an adaptation (by User:Ogrebot) of the original work of User:Bilou at en.wikipedia, published under a CC-BY-SA license, sourced from https://commons.wikimedia.org/wiki/File:Matrix_multiplication_diagram_2.svg

pathway, TNF-alpha to activate the TNF-alpha pathway as well as the NFkB pathway downstream of it, TGF-beta 1 to activate the TGFb pathway, and IFN-gamma to activate the JAK–STAT pathway. We subsequently measured phospho-proteomics (Fig. 2c) and gene expression upon perturbation. Results of these experiments confirmed that the PROGENy scores (Fig. 2d) capture pathway activity, as they accurately reflected the activated pathway and agreed with the measured changes in the activity status of key proteins in the corresponding pathways measured by phosphorylation.

Now that we have confirmed how pathway-responsive genes behave when a stimulus is present, we can take the idea one step further and hypothesize that the existence of a different basal expression level of the responsive genes may in turn correspond to cell-intrinsic signaling activity. When we apply PROGENy to a

cell line panel, we find that the obtained pathway scores are robust to changes in the experiments that the model was derived from (Fig. 2d), and also observe a similar correlation as the previously observed cross-activation upon perturbation (Supplementary Fig. 8).

**Recovering mechanisms of known driver mutations**. If our reasoning is correct and PROGENy signatures in basal gene expression correspond to intrinsic signaling activity, we should be able to see a higher pathway score in cancer patients with an activating driver mutation in that pathway and a lower score for pathway suppression compared to patients where no such alteration is present.

We selected all cancer types in the TCGA for which there were tissue-matched normals available, in order to make full use of the

pathway methods that require them. We calculated pathway scores for those using PROGENy, Reactome[9] and Gene Ontology[8] enrichment, SPIA[11], Pathifier[13], PARADIGM[12], a modified version of SPEED[5], and the Gatza et al.[18] signatures (Supplementary Table 2). We used an ANOVA to calculate significant associations between the presence and absence of mutations and copy number alterations and the inferred pathway scores for our method (Fig. 3a) and others (Supplementary Fig. 9).

In terms of proliferative signaling, we find that PROGENy identifies *EGFR* amplifications to activate both the EGFR and MAPK pathways (FDR < $10^{-9}$). *KRAS* mutations and amplifications show an increase in inferred MAPK/EGFR activity. Other methods do not detect a strong activation of the MAPK/EGFR pathways given those alterations (Fig. 3b.; top right and bottom left). We find the same effect for *BRAF* mutations (FDR < $10^{-10}$) that additionally activate TNFa/NFkB (FDR < $10^{-15}$).

For *TP53* mutations, PROGENy finds a significant reduction in p53/DNA damage response activity (FDR < $10^{-64}$) and activation of the PI3K and Hypoxia pathways (FDR < $10^{-15}$). This is in contrast to loss of *TP53*, where we only find a reduction in p53/DDR (FDR < $10^{-3}$), but no strong evidence of modification of any other pathway (FDR > 0.04). The dual nature of *TP53* mutations and loss are in line with the recent discovery that *TP53* mutations can act in an oncogenic manner in addition to disrupting its tumor suppressor activity, which has been shown for individual cancer types[28–31]. In addition, this analysis suggests a link between *TP53* mutations and genes that are induced by activation of canonical oncogenic signaling such as PI3K or the hypoxic response. Other methods (Fig. 3b.; top left) do not recover the expected negative association between these alterations and p53/

DDR activity. Gene Ontology showed a much weaker effect in the same direction, while Reactome, Pathifier, and SPIA showed an incorrect positive effect. These methods do, however, capture the activation of other oncogenic pathways, suggesting that this effect is driven by expression changes that then lead to changes in activity.

PROGENy finds that *VHL* mutations (which have a high overlap with Kidney Renal Carcinoma, KIRC) are associated with an expected stronger induction of hypoxic genes[32] compared to other cancer types (FDR < $10^{-200}$). It is the only method to recover hypoxia as the strongest link with *VHL* mutations, while the other methods primarily report expression changes in unrelated pathways (Fig. 3b.; bottom right). More surprisingly, we find that presence of *PIK3CA* amplifications and *PTEN* deletions is also more connected to increasing the hypoxic response (FDR < $10^{-6}$) compared to an effect on the PI3K-responsive genes (Supplementary Table 3). A role of PI3K signaling in hypoxia has been shown before[33–35].

These highlights reflect the more general pattern that PRO-GENy is able to correctly infer the impact of driver mutations that the other pathway expression-based methods could not. The latter are only able to identify some cases where activity is mediated by changes in the expression level of the pathway members itself.

**Associations with drug response.** The next question we tried to answer is how well PROGENy is able to explain drug sensitivity in cancer cell lines. We took as a measure of efficacy the IC$_{50}$, i.e., the drug concentration that reduces viability of cancer cells by 50%, for 265 drugs and 805 cell lines from the GDSC project[5]. We

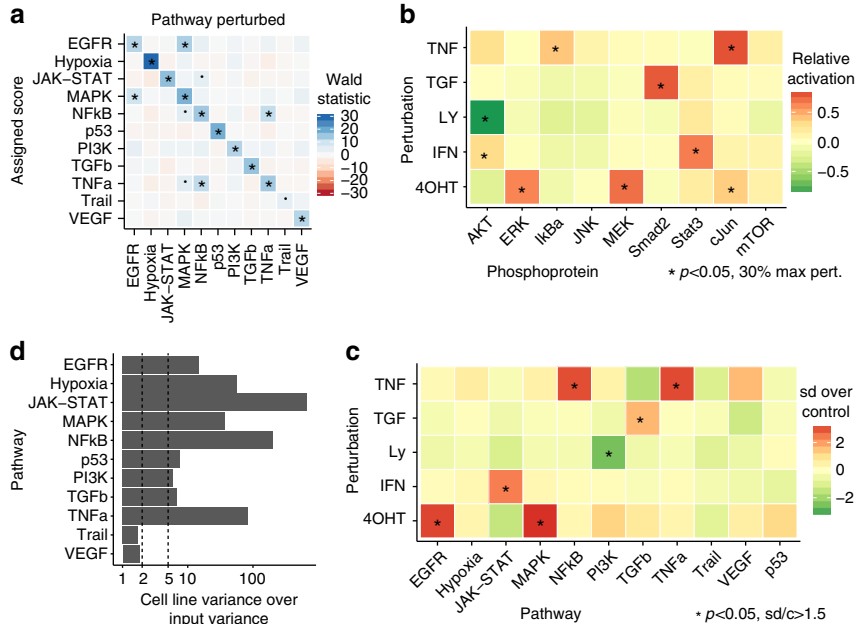

**Fig. 2** Evaluation of pathway-response signatures. **a** Associations for PROGENy pathway scores with experimental perturbation for experiments that the model was not built with (leave-one-out cross-validation). Each pathway is strongly associated with its own perturbation, and we observe few cases of cross-talk in agreement with biological knowledge. **b** Pathway perturbations in HEK293 cell line activate the corresponding signaling proteins. MEK and ERK for MAPK pathway, Stat3 for Interferon-induced JAK-STAT, AKT for PI3K, Smad2 for TGFb, and IKb for TNF-alpha-induced NFkB. As expected, all increased upon stimulation except AKT that decreased upon inhibition. Activation shown relative to maximum readout per antibody, p values reported for one-sample one-sided t test. Results are significant if p < 0.05 and perturbation is at least 30% of maximum. **c** PROGENy correctly infers pathway activity from gene expression in the HEK293 experiments. Associations are significant if p value of two-sample one-sided t test <0.05 and experiments are at least 1.5 standard deviations above or below the control. **d** Stability of basal pathway scores when bootstrapping input experiments. Bars show how much more variance in pathway scores (GDSC panel) is introduced by cell line identity over using resampled perturbation experiments in model building. Variance by cell line is over five times as high for most pathways, and roughly twice as high for Trail and VEGF

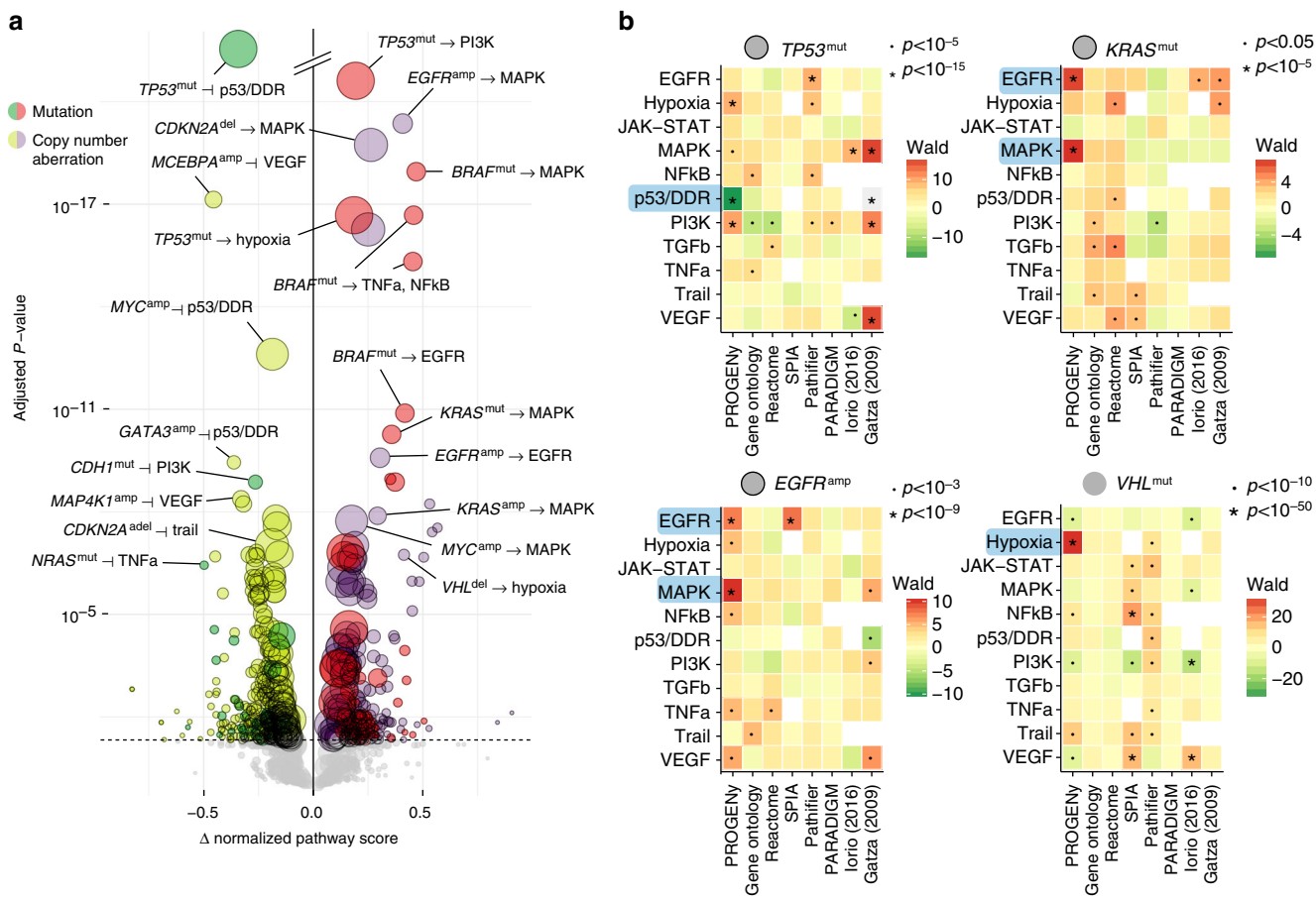

**Fig. 3** Ability of pathway methods to recover well-known mutations. **a** Volcano plot of pan-cancer associations between driver mutations and copy number aberrations with differences in pathway score. Pathway scores calculated from basal gene expression in the TCGA for primary tumors. Size of points corresponds to occurrence of aberration. Type of aberration is indicated by superscript "mut" if mutated and "amp"/"del" if amplified or deleted, with colors as indicated. Effect sizes on the horizontal axis larger than zero indicate pathway activation and smaller than zero indicate inferred inhibition. P values on the vertical axis FDR-adjusted with a significance threshold of 5%. Associations shown without correcting for different cancer types. Associations with a black outer ring are also significant if corrected. **b** Comparison of pathway scores (vertical axes) across different methods (horizontal axes) for *TP53* and *KRAS* mutations, *EGFR* amplifications and *VHL* mutations. Wald statistic shown as shades of green for downregulated and red for upregulated pathways. P value labels shown as indicated. White squares where a pathway was not available for a method

performed an ANOVA between those $IC_{50}$ values and inferred pathway scores of PROGENy and the other methods we investigated.

We found 178 significant associations for PROGENy (10% FDR in Fig. 4a and Supplementary Fig. 10), dominated by sensitivity associations between MAPK/EGFR activity and drugs targeting MAPK pathway (Fig. 4b) that are consistent with oncogene addiction. In particular, this includes associations of the MAPK/EGFR pathways with different MEK inhibitors (Trametinib, RDEA119, CI-1040, etc.), a RAF inhibitor (AZ628) and a TAK1 inhibitor (7-Oxozeaenol). However, the strongest hit we obtained was the association between Nutlin-3a and p53-responsive genes. Nutlin-3a is an MDM2-inhibitor that in turn stabilizes p53. Since it has also previously been shown that a mutation in *TP53* is strongly associated with increased resistance to Nutlin-3a[4], this is a well-understood mechanism of sensitivity (presence) or resistance (absence of p53 activity) to this drug that our method captures, but none of the pathway expression-based methods do.

Considering the overall number of associations, the other pathway methods provided a lower number across the range of significance (Fig. 4a). PROGENy outperforms associations obtained with driver mutations at 10% FDR, as those only yield 136 associations. The latter only provides stronger associations

for *TP53*, where the signature is a compound of p53 signaling and DNA damage response, and PLX4720/Dabrafenib, drugs that were specifically designed to target mutated *BRAF*. For 147 out of 265 drugs covered by significant associations with either PROGENy or driver mutations, PROGENy provided stronger associations for 78, with a significant enrichment in cytotoxic drugs compared to targeted drugs for mutations (Fisher's exact test, $p < 0.01$).

However, stratification using PROGENy and mutated driver genes is not mutually exclusive. Our pathway scores are able to further stratify the mutated and wild-type sub-populations into more and less sensitive cell lines (Fig. 4c and Supplementary Tables 4–5). This includes, but is not limited to, *BRAF*, *NRAS*, or *KRAS* mutations using MAPK pathway activity and the MEK inhibitor Trametinib (Fig. 4c; top left) or RAF inhibitor AZ628 (Fig. 4c; bottom left), *BRAF* mutations with Dabrafenib (Fig. 4c; top right), and *TP53* mutations with p53/DDR and Nutlin-3a (Fig. 4c; bottom left). For MAPK- and *BRAF*-mutated cell lines, we find that cell lines with an active MAPK pathway according to PROGENy are 65 (AZ628), 9130 (Trametinib), or $10^4$ fold (Dabrafenib) more sensitive than those where it is inactive. For Trametinib, cell lines with active MAPK, but no mutation in *BRAF*, *KRAS*, or *NRAS* are 15 times more sensitive than cell lines that harbor a mutation in any of them, but our analysis

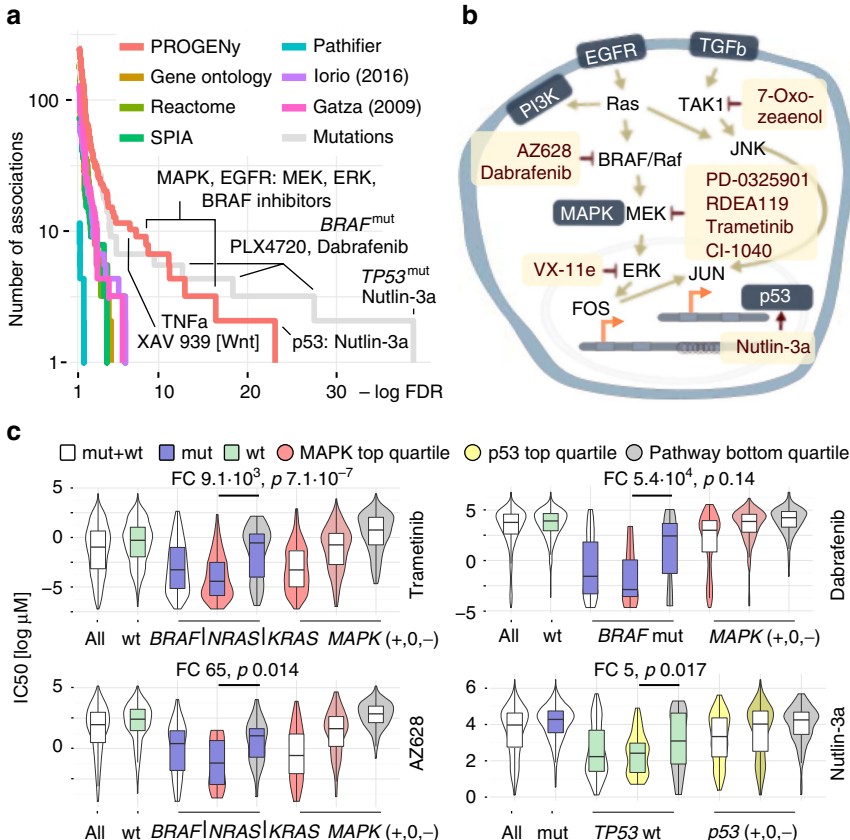

**Fig. 4** MAPK and p53 scores drive drug response across all cancer types. **a** Comparison of the associations obtained by different pathway methods. Number of associations on the vertical and FDR on the horizontal axis. PROGENy yield more and stronger associations than all other pathway methods. Mutation associations are only stronger for TP53/Nutlin-3a and drugs that were specifically designed to bind to a mutated protein. PARADIGM not shown because no associations <10% FDR. markers (green) and greater than zero resistance markers (red). P values FDR-corrected. **b** Pathway context of the strongest associations (Supplementary Fig. 10) between EGFR/MAPK pathways and their inhibitors obtained by PROGENy. **c** Comparison of stratification by mutations and pathway scores. MAPK pathway (*BRAF*, *NRAS*, or *KRAS*) mutations and Trametinib on top left panel, AZ628 bottom left, *BRAF* mutations and Dabrafenib top right, and p53 pathway/*TP53* mutations/Nutlin-3a bottom right. For each of the four cases, the leftmost violin plot shows the distribution of $IC_{50}$s across all cell lines, followed by a stratification in wild-type (green) and mutant cell lines (blue box). The three rightmost violin plots show stratification of all the cell lines by the top, the two middle, and the bottom quartile of inferred pathway score (indicated by shade of color). The two remaining violin plots in the middle show mutated (*BRAF*, *KRAS*, or *NRAS*; blue color) or wild-type (*TP53*; green color) cell lines stratified by the top- and bottom quartiles of MAPK or p53 pathways scores (Mann–Whitney U-test statistics as indicated)

determined that MAPK is inactive (Supplementary Table 5; fold changes reported for median of subset).

Taken together, these results show that PROGENy can be used to complement mutation-derived biomarkers by either refining them or providing an alternative where no such marker exists. Associations obtained with other methods do not show strong interactions between pathways and drugs that target their members (Supplementary Fig. 10). Furthermore, our associations hold true in an independent sensitivity screen for overlapping drugs (CCLE; Supplementary Table 6).

**Implications for patient survival.** The implications of inferred pathway activity compared to pathway expression is expected to be less clear for patient survival than for cell line drug response due to the many more factors that affect the phenotype observed. Nonetheless, we were interested in how our inferred pathway activity compared to pathway expression methods in terms of overall patient survival.

Across all cancer types, PROGENy found a strong association between the activation of EGFR, MAPK, PI3K, and Hypoxia pathways and decreased survival, similar to other signature methods (Fig. 5a). Gene Ontology found much weaker

associations for expression of those pathways, and the other pathway mapping methods missed them almost entirely. PROGENy is the only method to find an increase in survival associated with the activation of the Trail/apoptosis pathway, while other methods show either a decrease or no effect, or we did not find an appropriate pathway in the signatures we compared. For JAK–STAT, NFkB, p53, and VEGF pathways there are no significant associations that are picked up by more than one method (FDR < 0.05). Compared to pathway mapping, signature methods provide associations of similar strength with overlapping pathways.

For individual cancer types, PROGENy finds a similar separation between oncogenic and tumor-suppressor pathways (Fig. 5b), showing that it can capture both general and specific patterns in gene expression changes. Importantly, pathway mapping methods do not provide this separation and our associations are significant for more cancer types and more specific to individual pathways (Supplementary Fig. 11). In addition, we find cancer-specific associations of pathways with no effect in the pan-cancer setting: For instance, with PROGENy, Adrenocortical Carcinoma (ACC) shows a significant increase of survival with p53 activity (FDR < $10^{-3}$). This positive effect of p53 on survival is supported by the fact that ACC samples do not

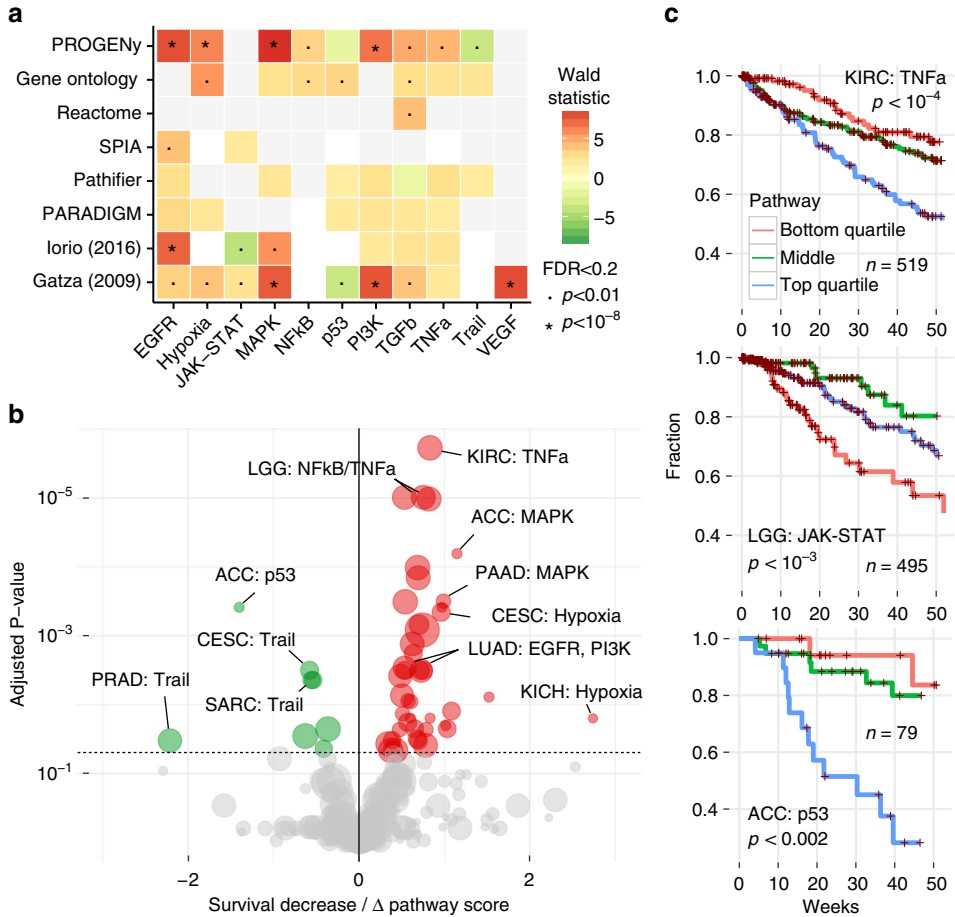

**Fig. 5** Response signatures outperform pathway methods for patient survival. **a** Pan-cancer associations between pathway scores and patient survival. Pathways on the horizontal axis and different methods on the vertical axis. Associations of survival increase (green) and decrease. Significance labels as indicated. Shades correspond to effect size, p values as indicated. **b** Volcano plot of cancers-specific associations between patient survival and inferred pathway score using PROGENy. Effect size on the horizontal axis. Below zero indicates increased survival (green), above decreased survival (red). FDR-adjusted p values on the vertical axis. Size of the dots corresponds to number of patients in each cohort. **c** Kaplan–Meier curves of individual associations for kidney (KIRC), low-grade glioma (LGG), and adrenocortical carcinoma (ACC). Pathway scores are split in top and bottom quartiles and center half. Lines show the fraction of patients (vertical axis) that are alive at a given time (horizontal axis) within one year. P values for discretized scores

harbor any previously reported gain-of-function *TP53* variants[31]. Kidney Renal Clear Cell Carcinoma (KIRC) and Low-Grade Glioma (LGG) show decreased survival with TNFa and JAK-STAT pathways, respectively, where specific activating mutations are much less known than for EGFR/MAPK. For these three associations, the top and bottom quartiles of PROGENy pathway activity were able to stratify patients in groups with over 25% difference in one year survival (Fig. 5c). These associations are stable when resampling patients (Supplementary Table 7).

In summary, we can observe that signature-based methods generally outperform pathway mapping for survival associations, but the difference between PROGENy and the signatures of SPEED[5,26] and Gatza et al.[18] is less pronounced than for driver mutations or drug response.

## Discussion

The explanation of phenotypes in cancer, such as cell line drug response or patient survival, has largely been focussed on genomic alterations (mutations, copy number alterations, and structural variations). While this approach has generated many important insights into cancer biology, it does not directly make statements about the impact of those aberrations on cellular processes and signal transduction in particular. Pathway methods, mostly used on gene expression, have produced mixed results

when it comes to delivering actionable evidence. This can in part be due to lack of robustness, as suggested by the heterogeneity in responses of individual signatures (Supplementary Fig. 6), but arguably also by the fact that extracting features that reflect pathway activity from gene expression is not trivial. With proteomics lagging behind sequencing data for the foreseeable future, we have a need to address the accurate inference of pathway activity from gene expression in heterogeneous samples using a general downstream gene expression pattern.

We developed PROGENy in order to achieve this. PROGENy leverages a large compendium of pathway-responsive gene signatures derived from a wide range of different conditions in order to identify genes that are consistently deregulated. While this approach has been taken before, previous studies either focussed less on integrating responses from many different cell lines[25] or derived their scores from a much smaller collection of perturbation experiments[5,26].

We found that despite the heterogeneity of individual gene expression experiments, PROGENy closely corresponds to pathway perturbations. PROGENy can recover the impact of known driver mutations from basal-gene expression, but also identify cases where a pathway is active without their presence. In contrast, pathway mapping only recovers known associations, where this effect is mediated by expression changes in pathway

members, such as *TP53* oncogene activation or copy number aberrations. We applied PROGENy to a drug sensitivity data set, where the significant associations we obtained corresponded better to known drug–pathway interactions than those of competing methods. PROGENy was also able to consistently distinguish between oncogenic driver pathways (mainly EGFR and MAPK) and cell death (Trail) pathways for patient survival.

Overall, our results suggest that PROGENy provides a better measure of pathway activity than other pathway methods, irrespective of whether the latter was derived from gene sets or directed paths. The latter can be used for many more pathways, as information on the pathway components is more often available than perturbation experiments. However, our results indicate that one should be cautious when interpreting the expression level of a pathway as its activity.

We have shown that PROGENy is able to refine our understanding of the impact of mutations, as well as their utility for cell line drug response and patient survival. It provides a strong evidence that in order to infer pathway activity, e.g., for patient stratification, a downstream readout should be used instead of mapping transcript expression levels to signaling molecules.

While PROGENy provides a good estimate of pathway activity in large and heterogeneous data sets, signatures derived from, for instance, a specific tissue may still more closely reflect activation status given the same context. We see a hint of this when applying the Gatza et al. signatures for the TCGA breast cancer cohort, but more studies will be required to fully elucidate the differences between a common response and additional transcriptional modules that may not always be activated. We believe that our curated set of experiments and computational pipeline will be useful to further investigate this aspect of specialized vs. consensus signatures and when either of them should be used.

## Methods

**Data from The Cancer Genome Atlas (TCGA).** To obtain the TCGA data, we used the Firehose tool from the BROAD institute (http://gdac.broadinstitute.org/), release 28 January 2016.

For gene expression, we used all data labeled "Level 3 RNA-seq v2". We extracted the raw counts from the text files for each gene, discarded those that did not have a valid HGNC symbol, and averaged expression levels where more than one row corresponded to a given gene. We then performed a variance stabilizing transformation (*DESeq2* package[36], BioConductor) for each TCGA study separately, to be able to use linear modeling techniques with the count-based RNA-seq data. The data used corresponds to 34 cancer types and a total of 9737 tumor and 641 matched normal samples.

From the clinical data, we extracted the vital status and used known survival time or known time of last follow-up as the survival time for the downstream analyses. We converted the time in days to months by dividing by 30.4. We obtained both mRNA expression levels as well as survival times for 10,544 patients, distributed across 33 cancer types. For comparing different pathway methods, we only used cancer types with tissue-matched controls, leaving 5927 samples in 13 cancer types.

**Data for cell line gene expression and drug sensitivity.** We used version 17a of the Genomics of Drug Sensitivity in Cancer (GDSC) data[5], comprised of molecular data for 1001 cell lines and 265 anticancer drugs, specifically microarray gene expression data (ArrayExpress accession E-MTAB-3610) and the IC$_{50}$ values for each drug–cell line combination. For computing pan-cancer associations, we used the subset with TCGA-like cancer type label, leaving 805 cell lines.

We downloaded the Cancer Cell Line Encyclopedia (CCLE) microarray gene expression and drug sensitivity data from the CCLE web page (https://portals. broadinstitute.org/ccle). For microarray data (2013–03–18), we performed RMA normalization, and mapped the probes to HGNC gene symbols. We used drug profiling data version 2012–02–20 and drug metadata version 2015–02–24.

**Perturbation experiments of HEK293 cell line.** HEK293ΔRAF1:ER cells were acquired and cultured as previously described[37]. Before treatments, cells were starved in serum-free medium overnight. Cells were treated with 4-hydroxy tamoxifen (4OHT, Sigma-Aldrich; 0.5 μM), Ly294002 (Life Technologies; 10 μM) or the following ligands from Peprotech: TNF-alpha (20 ng/ml), TGF-beta 1 (10 ng/ml), IFN-gamma (50 ng/ml). Cell lines have been tested for Mycoplasma infection using Tenor GeM Classic (Minerva Biolabs).

**RNA sequencing for HEK293 perturbations.** After 4 h of treatment, total RNA was extracted with Qiagen RNeasyMini Kit. Sequencing libraries were prepared using Illumina TruSeq mRNA Library Prep Kit v2 and sequenced on Illumina HiSeq 2000. Read quality was assessed using FastQC and sequencing adapters were trimmed using cutadapt[38]. Reads were mapped with STAR aligner v2.5.0c[39] on hg19 using GENCODE v19 for annotation and quantified with subread feature-Counts[40]. The preprocessing pipeline was written in Snakemake[41]. Raw read counts were then normalized with DESeq2 and variance stabilization transformed[36].

**Phosphoprotein measurements for HEK293 perturbations.** Protein extracts of cells were prepared by incubation with cell lysis buffer (Bio-Plex Pro Cell signaling Reagent Kit, Bio-Rad). The Bio-Plex Protein Array system (Bio-Rad, Hercules, CA) was used, as described earlier[42]. A total of 10 μg protein was analyzed. The following analytes were used: AKTS473, c-JunS63, ERK1/2T202,Y204/T185,Y187, IkBaS32,S36, JNKT183,Y185, MEK1S217,S221 and mTORS2448. The beads and detection antibodies were diluted 1:3. For data acquisition, the xPONENT software was used.

The following antibodies were used for western blot measurements: rabbit anti human p-SMAD2 (Ser465/467) (138D4) #3108, rabbit anti human p-Stat3 (Tyr 705) #9131 and rabbit anti human ß-Tubulin #2146. All primary antibodies were diluted 1:1000 and obtained from Cell Signaling Technology. Electrophoresis was performed and lysates were transferred onto nitrocellulose membranes. Unbound protein sites were blocked with 1:2 Odyssey Blocking Buffer (from Li-COR) and PBS. Thereafter, specific proteins were detected by incubation with primary antibodies diluted in the same blocking buffer containing 0.1% Tween-20 overnight at 4 °C followed by near-infrared dye labeled secondary antibodies. For detection of phosphorylated SMAD2 and Stat3, a total of 30 and 60 μg protein was used, respectively. Membrane Images were taken using Li-COR Odyssey Fc. The bands were quantified by determining the background corrected total intensities using ImageStudio software (Li-COR). All Signals were normalized to ß-Tubulin.

Two biological replicates were measured both after 30 min and 1 h and outcomes were analyzed together by calculating log2 ratios to their respective solvent control (BSA).

**Curation of perturbation-response experiments.** Our method is dependent on a sufficiently large number of available perturbation experiments that activate or inhibit one of the pathways we were looking at. The following conditions needed to be met in order for us to consider an experiment: (1) the compound or factor used for perturbation was one of our curated list of pathway-perturbing agents (Supplementary Note 1); (2) the perturbation lasted for less than 24 h to capture genes that belong to the primary response; (3) there was raw data available for at least two control arrays and one perturbed array; (4) it was a single-channel array; (5) we could process the arrays using available BioConductor packages; (6) the array was not custom-made so we could use standard annotations.

We curated a list of known pathway activators and inhibitors for 11 pathways, where the interaction between each compound and pathway is well established in literature (Supplementary Note 1). We then used those as query terms for public perturbation experiments in the ArrayExpress database[43] and included a total of 219 submissions and 581 experiments in our data set, where each experiment is a distinct comparison between basal and perturbed arrays. If there were multiple time points, different cells, different concentrations, or different perturbing agents within a single database submission, they were considered as different experiments.

**Microarray processing.** Started from the curated list of perturbation-induced gene expression experiments, we included all single-channel microarrays with at least two replicates in the basal condition with raw data available that could be processed by either the limma[44], oligo[45], or affy[46] BioConductor packages and for which there was a respective annotation package available.

We first calculated a probe-level expression levels for 581 full series of arrays, where we performed quality control of the raw data using RLE and NUSE cutoffs under 0.1 and kept all arrays below that threshold. If after filtering less than two basal condition arrays remained, the whole experiment was discarded. For the remaining 575 experiments we normalized expression data using the RMA algorithm and mapped the probe identifiers to HGNC symbols.

**Building a linear model of pathway-response genes.** We set aside 10 experiments for model validation. For the remainder and each HGNC symbol, we calculated a model based on mean and standard deviation of the gene expression level, and computed the z-score as average number of standard deviations that the expression level in the perturbed array was shifted from the basal arrays. We then performed LOESS smoothing for all z-scores in a given experiment using our null model as described previously[26]

From the z-scores of all experiments and all pathways, we performed a linear regression with the pathway as input and the z-scores as response variable for each gene separately:

$$Z_g \sim M_\mathrm{p} \dots \forall g, \mathrm{p}$$

Where $Z_g$ is the z-score for a given gene g across all input experiments. $M_p$ is a perturbation indicator vector across all input experiments for each pathway p that has the coefficient 1 if the experiment had a pathway activated, −1 if inhibited, and 0 otherwise. For instance, the Hypoxia pathway had experiments with low oxygen conditions set as 1, HIF1A knockdown as −1, and all other experiments as 0. The same is true for EGFR and EGF treatment vs. EGFR inhibitors, respectively, with the additional coefficients of MAPK pathways set to 1 because of known cross-talk. TNFa perturbations also changed NFkB coefficients for the same reason.

From the result of the linear model, we selected the top 100 genes per pathway according to their significance (pvalue) and took their estimate (the fitted z-scores) as coefficient. We set all other gene coefficients to 0. This way, we obtained a matrix with HGNC symbols in rows and pathways in columns, where each pathway had 100 non-zero gene coefficients (Supplementary Data 1).

**PROGENy scores.** Each column in the matrix of perturbation-response genes corresponds to a plane in gene expression space, in which each cell line or tumor sample is located. If you follow its normal vector from the origin, the distance it spans corresponds to the pathway score P, each sample is assigned (matrix of samples in rows, pathways in columns). In practice, this is achieved by a simple matrix multiplication between the gene expression matrix (samples in rows, genes in columns, values are expression levels) and the model matrix (genes in rows, pathways in columns, values are our top 100 coefficients):

$$P = E \times G$$

We then scaled each pathway or gene set score to have a mean of zero and standard deviation of one, in order to factor out the difference in strength of gene expression signatures and thus be able to compare the relative scores across pathways and samples at the same time.

**EPSA model.** The EPSA model was built as previously published[25] with the following modifications: (1) we used the mean of the treated and untreated arrays for each experiment in order to avoid bias by experiment size; (2) we calculated significance of differential expression with limma[44], not SAM; and (3) we selected the top 100 significant genes due to very different gene numbers at 5 or 10% FDR.

**Comparison to other signature and signature consensus methods.** We calculate pathway scores for all perturbation experiments in the direction of activation (activated—control and control—inhibition). For methods that work on differential expression (SPEED: both using the original web server at https://speed.sys-bio.net and running the method on our perturbation experiments, GSEA using Kolmogorov–Smirnov statistic), we use the negative logarithm of the p value as pathway score. For methods that score individual samples (PROGENy, EPSA), we use the difference of the mean between basal and perturbed arrays for each experiment. For both, we normalize the pathway scores per experiment because of the different strength of perturbations. We then quantify how well each pathway signature ranks experiments, where a pathway was perturbed before experiments where a pathway was not perturbed by the Receiver Operator AUC. We quantify if a given method has a consistently higher AUC than another across pathways using a binomial test (Supplementary Fig. 6a).

In addition, we quantify the influence of the number of signatures on the ROC AUC. For this, we build the PROGENy model by sampling n signatures per pathway 10 times with replacement and calculate the AUC as described above (Supplementary Fig. 6b).

**Validation of PROGENy scores on public experiments.** We previously set aside 10 public perturbations experiments that measure both pathway activation (mainly western Blots) and gene expression upon perturbation, which were not included in any of the model building. For each of those experiments, we quantified the Blot bands in the original publication (DOI and experimental details in Supplementary Fig. 7) using ImageJ for the control vs. perturbed condition if no numerical values were reported. We calculated PROGENy pathway scores for both the control and perturbed condition, and plotted the spread of the control scores vs. the spread of the perturbed scores. We set the median of the control to 0, and the total standard deviation of the control-perturbed pair to 1 for easier presentation (without changing test statistics). We performed a one-tailed t test between each control and perturbed pair and report the p values (Supplementary Fig. 7).

**Validation of PROGENy scores on HEK293 perturbations.** We confirmed pathway activation using MEK for the MAPK pathway, Stat3 for JAK-STAT, AKT for PI3K, Smad2 for TGFbeta, IKb for NFkB by performing a one-sample one-tailed t test of the fold change over BSA, including samples from both 0.5 and 1 h after perturbation. We report all fold changes with a p value <0.05 and at least 30% of the maximum antibody readout as significant (Fig. 2c). We then computed the pathway scores for all conditions, and scaled each pathway score to have a mean of 0 and standard deviation of 1. We then computed the difference between the control condition (BSA treatment) and each perturbation. For this comparison, we plot the difference in means (Fig. 2c) and perform a one-tailed t test. Here, we

reported all pathway changes as significant if they have a p value <0.05 and an activation status that is 1.5 standard deviations above or below the control.

**Pathway scores using gene sets.** We matched our defined set of pathways to the publicly available pathway database Reactome[9] and Gene Ontology (GO)[8] categories, as well as Gatza et al. signatures (Supplementary Table 2a, b, f), to obtain a uniform set across pathway resources that makes them comparable. The SPEED platform already uses the same pathways, so no mapping was required. We calculated pathway scores as Gene Set Variation Analysis (GSVA) scores that are able to assign a score to each individual sample (unlike GSEA that compares groups).

**SPIA scores.** Signaling Pathway Impact Analysis (SPIA)[11] is a method that utilizes the directionality and signs in a KEGG[47] pathway graph to determine if in a given pathway structure the available species are more or less available to transduce a signal. As the species considered for a pathway are usually mRNAs of genes, this method infers signaling activity by the proxy of gene expression. In order to do this, SPIA scores require the comparison with a normal condition in order to compute both their scores and their significance.

We used the SPIA Bioconductor package[11] in our analyses, focussing on the subset of pathways used by the other methods (Supplementary Table 2c). We calculated our scores either for each cell line compared to the rest of a given tissue where no normals are available (i.e. for the GDSC and drug response data) or compared to the tissue-matched normals (for the TCGA data used in driver and survival associations).

**Pathifier scores.** As Pathifier[13] requires the comparison with a baseline condition in order to compute scores, we computed the GDSC/TCGA scores as with SPIA. As gene sets, we selected Reactome pathways that corresponded to our set of pathways (Supplementary Table 2b), where Pathifier calculated the "signal flow" from the baseline and compared it to each sample.

**PARADIGM scores.** We used the PARADIGM software from the public software repository (https://github.com/sbenz/Paradigm) and a model of the cell signaling network[48] from the corresponding TCGA publication (https://tcga-data.nci.nih.gov/docs/publications/coadread_2012/). We normalized our gene expression data from both GDSC and TCGA using ranks to assign equally spaced values between 0 and 1 for each sample within a given tissue. We then ran PARADIGM inference using the same options as in the above publication for each sample separately. We used nodes in the network representing pathway activity to our set of pathways (Supplementary Table 2d) to obtain pathway scores that are comparable to the other methods, averaging scores where there were more than one for a given sample and node.

**Recall of perturbation experiments.** We calculate pathway scores for each of our curated experiments using all pathway methods. For gene set methods (GO, Reactome, Gatza) we use the difference in GSVA without kernel density estimator due to low sample numbers. For PROGENy, we exclude the experiment we quantify from model building (leave-one-out cross-validation).

We calculate linear associations between perturbations and assigned scores and plot the assigned pathway scores (using whether a pathway was perturbed as 0/1 coefficients and the pathway scores as response variable; pathway inhibitions encoded as negative pathway activations) and show the relative (column scale function in pheatmap) activation patterns as heatmaps (Supplementary Fig. 7).

**Associations with known driver mutations and CNAs.** For comparing the impact of mutations across different pathway methods, we used TCGA cohorts, where tissue-matched controls were available, leaving 6549 samples across 13 cancer types. For mutated genes, we considered all genes that had a change of coding sequence (SNP, small indels in MAF files) as mutated and all others as not mutated. For copy number alterations (CNAs), we used the thresholded GISTIC[49] scores, where we considered homozygous deletions (−2) and strong amplifications (2) as altered, no change (0) as basal and discarded intermediate values (−1, 1) from our analysis. We focussed our analysis of the mutations and copy number alterations on the subset of 464 driver genes that were also used in the GDSC. We used the sets of mutations and CNAs to compute the linear associations between samples for all different methods we looked at.

**Drug associations using GDSC cell lines.** We performed drug association using an ANOVA between 265 drug $IC_{50}$s and 11 inferred pathway scores conditioned on MSI status, doing a total of 2915 comparisons for which we correct the p values using the False Discovery Rate. For pan-cancer associations, we used the cancer type as a covariate in order to discard the effect that different tissues have on the observed drug response. While this will also remove genuine differences in pathway activation between different cancer types, we would not be able to distinguish between those and other confounders that impact the sensitivity of a certain cell line from a given tissue to a drug. Our pan-cancer association are thus the same of intra-tissue differences in drug response explained by inferred (our method, GO, or Reactome) pathway scores.

We selected four of our strongest associations to investigate whether they provide additional information of what is known by mutation data. For two MEK inhibitors, we show the difference between wild-type and mutant MAPK pathway (defined as a mutation in either *NRAS*, *KRAS*, or *BRAF*) with a discretized pathway score (upper and lower quartile vs. the rest), as well as the combination between the upper quartile of tissue-specific pathway scores and presence of a MAPK mutation. For a BRAF inhibitor, we show additional stratification on top of *BRAF* mutations, and for Nutlin-3a on top of *TP53* mutations.

**Survival associations using TCGA data.** Starting from the pathway scores derived with GO/Reactome GSEA, SPIA, Pathifier, PARADIGM, and our method on the TCGA data as described above, we used Cox Proportional Hazard model (R package *survival*) to calculate survival associations for pan-cancer and each tissue-specific cohort. For the pan-cancer cohort, we fitted the model for each pathway and method separately, regressing out the study of origin and age of the patient. For the tissue-specific cohorts, we regressed out the age of the patients. We adjusted the *p* values using the FDR method for each method and study separately. We selected a significance threshold of 5 and 10% for the pan-cancer and cancer-specific associations for which we show a matrix plot and a volcano plot of associations, respectively.

In order to get distinct classes needed for interpretable Kaplan–Meier survival curves (Fig. 4c), we split three of our obtained pathway scores in upper, the two middle, and lower quartile.

**Code availability.** *progeny* is available as an R package on Bioconductor. The code used for the analysis in this paper is available at https://github.com/saezlab/footprints.

**Data availability.** RNA-Seq data are accessible from gene expression omnibus (GEO) under accession number GSE97979. Phosphoprotein measurements are available as Supplementary Data 2.

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

## Acknowledgements

M.S. is funded by a MRC Case fellowship (1246915) awarded to J.S.-R. and Joanna Betts (GSK). N.B. acknowledges funding by BMBF (OncoPath). M.J.G. is supported with funding from the Wellcome Trust (102696), Stand Up To Cancer (SU2C-AACR-DT1213), The Dutch Cancer Society (H1/2014-6919) and Cancer Research UK (C44943/A22536). We thank Francesco Iorio, Florian Markowetz, Bence Szalai and Alvis Brazma for useful discussions. We thank S. Cagnol and P. Lenormand for providing the HEK293ΔRAF1:ER cell line.

## Author contributions

M.S. designed research, performed all analyses, and wrote the manuscript. A.S., F.U., B.K. and S.S. performed and preprocessed validation experiments, supervised by N.B. B.K., M. K., N.B. and M.J.G. supported result interpretation and manuscript writing. J.S.-R. supervised the project and contributed to writing the manuscript.

## Additional information

**Competing interests:** The authors declare no competing financial interests.

