## [Peer Review File · Nature Communications]

Reviewers' comments:

Reviewer #1 (Remarks to the Author):

Whilst the PRODGENTY method clearly outperforms the other methods (Reactome, GO, Pathifier, SPIA etc.) it is not experimentally validated nor exhaustively validated computationally. The authors utilize published perturbation data and response data (drugs sensitivity, patient outcome) but there is no direct experimental validation. This leads to some major concerns:

1. The study claims to solve the problem of gene expression data not reflecting PTMs, however this is not validated in the manuscript. There is no direct evidence provided that gene expression even when generated in response to a perturbation (e.g. EGF stimulation) provide insight that capture PTMs or signaling states. This is the major claim of the work, but it is not validated or proven in the current manuscript.

2. Related to this problem is the use of static pathway maps. These maps are inherently incomplete and with errors as the physical interactions comprising them are dynamically changed in response to perturbations, PTMs, tissues, cell lines. Thus the starting point for the PRODGENTY method is not reflective of the actual networks present at the time of stimulation, it is unclear how the method deals with this problem and how the networks highlighted are indeed causal/mechanistic in nature. They are found to be associative and predictive but it is not clear this infers causality. Related to point below.

3. The authors do not provide experimental evidence that the identified patterns are causal and real mechanisms underlying e.g. drug/patient response/outcome. The authors identify correlations but it is not clear these are causal nor that they are real as no direct validation is conducted.

Reviewer #2 (Remarks to the Author):

This is a very interesting paper that describes a simple yet apparently effective method for deriving pathway activity scores from RNA expression data. The authors apply this method to prediction of the effect of cancer driver mutations on pathway activities, to the prediction of the effects of small molecules on pathway activities, and to stratification of cancer patients by their survival. The authors benchmark their method against several other widely used methods and show a marked improvement in sensitivity and accuracy. They are also able to show that the differential response of cell lines to drugs can be explained in some cases by the predicted effects of cell line specific mutations. Potentially, this method will find multiple research applications, most importantly the identification of drug response biomarkers for precision medicine.

My enthusiasm for this paper is tempered by difficulties understanding some of the key assertions due to poor writing. There are numerous examples of this. I will pick out a few that stood out:

"These responsive genes are specific to the perturbed pathway (Supplementary Fig. 3) and do almost not overlap with genes that comprise it (Supplementary Fig. 4)" This sentence doesn't parse, and the easiest interpretation, which is that genes that respond to a pathway perturbation are usually not contained in the pathway, is hard to comprehend.

Another example:

"Adrenocortical Carcinoma (ACC) shows a significant survival increase with p53 activity (FDR<10⁻³), supported by the fact that it not harbor any previously reported 6 gainof-function TP53 variants."

And another:

"For the pancancer cohort, we regressed out the effect of the study and age of the patient, and fitted the more for each pathway and method used."

I would like to see a resubmission of this paper after a careful re-read by a native English speaker and correction of these

Methodologically, the main problem that occurred to me during review is that the testing data sets from GDSC and TCGA have previously been submitted to ArrayExpress and GEO and may have contaminated the training set. However, review of the supplementary materials revealed that this is not the case. A statement to this effect would help future readers.

When comparing PROGENy to other pathway-based activity prediction methods, it would be informative to directly describe the overlap between the genes that define a pathway set in each of the other methods (example, the EGFR Reactome genes) and those that have the strongest coefficients in the PROGENy linear regression. It is likely that the most of the differences in predictive power have to do with the fact that curated pathways focus on post-translational events and are relatively weak with respect to describing downstream transcriptional effects.

Reviewer #3 (Remarks to the Author):

The paper touches upon important point, which is the usefulness of gene expression signatures learned from perturbation experiments, and specifically of focusing on the core common to many different contexts. These points are not novel, and very similar approaches have been reported before, but the study is comprehensive, the data available today allows more accurate signatures than before, and the authors apply it to several interesting analyses, showing relevance to drug response and patient prognosis. If the points below are addressed well, I think the paper should be considered for publication.

Major remarks:

1. The authors compare their method, which is based on learning gene signatures from experimental data, to general pathway analysis approaches, which relies only on the knowledge of the pathway and does not require many controlled experiments (nor are limited to signaling pathways which this paper focus on). While they mention that others have tried to learn signatures from perturbations before, they make it sound as if it was limited to breast cancer and proven useless, where in fact these have been widely used successfully in many contexts (such as PMID: 18937865, PMID: 17008526, PMID: 27098033, PMID: 22549044, PMID: 20215513, PMID: 20591134, PMID: 18652687 and many more). This should be better reflected in the text, and in addition the authors should compare their method to these approaches, and not only to methods of applying generic genes sets and pathways. If they want also to compare to such generic methods, they should first clearly state the difference in approaches, their advantages and disadvantages, and also use these methods to deduce pathway scores of the gene sets they learned from the perturbation data, otherwise it is actually mostly a comparison of the Reactome/KEGG gene set to their gene set.
2. The novelty of the research may lie in the claim of a robust method to infer a common core of the pathways they study. As such, they should validate this claim in a more rigorous manner. One thing that is clearly missing is validating the signatures on a validation set not used to learn these signatures. A deeper look into the biological meaning of the key genes in their signatures, showing that many of them correspond to known core members of these pathways will also help support this claim. More generally all the analyses done on the same data used to derive the signatures (e.g. the results shown in Figure 2) should be done on an independent validation set. Similarly, the

Cox proportional hazard model should be learned on a training set and applied on a disjoint validation set.

We thank the reviewers for their comments, which we believe helped us to improve the content and clarity of the manuscript.

A major concern was the experimental validation of the signatures. To address this point, we performed two analyses to experimentally validate our pathway signatures obtained from gene expression to orthogonal measures of pathway activation from phosphorylation experiments. First, we used public data where both gene expression and phosphorylation experiments were available upon pathway stimulation (Supplementary Fig. 5). Second, we performed novel experiments with a cell line (HEK293) where we measured both phosphorylation and gene expression after stimulation of the pathway (Fig. 2c, Fig. 2d). In both public and our own data, we found a very good agreement between the scores obtained with PROGENy from gene expression with phosphorylation markers of signaling, hence providing experimental validation to our signatures. These new experiments were performed by Anja Sieber, Florian Uhlitz, Sascha Sauer, who are now added as coauthors.

The other major additions to our manuscript are that we now (i) assess PROGENy perturbation scores (Fig. 2a and Supplementary Fig. 6) in a leave-one-out manner; (ii) validate our drug associations in the CCLE (Supplementary Table 7), as well as the survival associations by bootstrapping the TCGA patients (Supplementary Table 8), and (iii) extend the comparison to other methods to include two other signature methods ('Iorio et al. 2016', and 'Gatza et al., 2009'; Figs. 3-5).

Reviewer #1 (Remarks to the Author):

Whilst the PROGENy method clearly outperforms the other methods (Reactome, GO, Pathifier, SPIA etc.) it is not experimentally validated nor exhaustively validated computationally. The authors utilize published perturbation data and response data (drugs sensitivity, patient outcome) but there is no direct experimental validation. This leads to some major concerns:

1. The study claims to solve the problem of gene expression data not reflecting PTMs, however this is not validated in the manuscript. There is no direct evidence provided that gene expression even when generated in response to a perturbation (e.g. EGF stimulation) provide insight that capture PTMs or signaling states. This is the major claim of the work, but it is not validated or proven in the current manuscript.

We agree with the reviewer that it is important to show how our gene expression signatures capture PTMs or signaling states. We have performed two sets of analyses to investigate this point. First, we have reviewed the literature and publicly available data for cases where for the same stimulation signaling activity (typically via phosphorylation) and gene expression changes are measured. We used PROGENy signatures (computed without these experiments) to estimate pathway from expression, and compared these with the signaling values. The results (Supplementary Fig. 7) confirm the agreement between the PTMs and gene expression signatures for all pathways we studied.

To further validate our signatures, we performed novel experiments with the HEK293 cell line. We stimulated them with five different ligands and inhibitors, and we measured

changes in phosphorylation of key proteins and gene expression changes. From the gene expression we computed the PROGENY scores, that agreed with the stimulation used and the phosphorylation measurements (Fig. 2c, 2d).

With these two additional analyses, we verify that our signatures agree with phosphoproteomic measurements when activating the corresponding pathways.

2. Related to this problem is the use of static pathway maps. These maps are inherently incomplete and with errors as the physical interactions comprising them are dynamically changed in response to perturbations, PTMs, tissues, cell lines. Thus the starting point for the PROGENY method is not reflective of the actual networks present at the time of stimulation, it is unclear how the method deals with this problem and how the networks highlighted are indeed causal/mechanistic in nature. They are found to be associative and predictive but it is not clear this infers causality. Related to point below.

We agree with the reviewer that pathway maps are incomplete and unable to capture per se dynamic and cell-specific aspects of signal transduction. To circumvent this limitation, our method does not explicitly model pathway structure and activation dynamics. Rather, we solely rely on the link between a known perturbation of a pathway and the effect on gene expression it mediates. This is in line with how gene expression signatures generally work and are used in large-scale projects such as the Connectivity Map.

It is also true that while our model reflects gene expression changes caused by pathway activation or inhibition, it does not immediately follow that these changes are present and detectable as intrinsic steady state footprints of signaling activity. To investigate the causality of the signatures, we used well established cancer driver mutations in an independent dataset (cancer patient data from the TCGA; Fig. 3) showing that when key pathway components are mutated and thereby the pathway hyper- or deactivated, the signatures change accordingly. We confirm this with independent activity and gene expression measurements upon perturbation (Fig. 2c, Fig. 2d and Supplementary Fig. 7; also mentioned above).

3. The authors do not provide experimental evidence that the identified patterns are causal and real mechanisms underlying e.g. drug/patient response/outcome. The authors identifies correlations but it is not clear these are causal nor that they are real as no direct validation is conducted.

The reviewer is correct that there is no causal association between the pathway activations and drug/patient response/outcome. What we do believe is (a) that pathway perturbations are causal to gene expression changes and (b) that those gene expression changes can be used to infer upstream signaling activity.

In response to this, we added evidence of a causal link between each pathway modulator and its effect on the corresponding pathway activity (Supplementary Note 2).

Biomarkers are correlative per se. As such, they are usually evaluated by deriving them from one data set and evaluating them in another data set. We derive them from perturbation experiments that are independent of TCGA cancer drivers, differential drug response due to

oncogene addiction and differential survival for oncogenic vs. tumor suppressor pathways
(cf. Fig. 3-5).

Reviewer #2 (Remarks to the Author):

This is a very interesting paper that describes a simple yet apparently effective method for deriving pathway activity scores from RNA expression data. The authors apply this method to prediction of the effect of cancer driver mutations on pathway activities, to the prediction of the effects of small molecules on pathway activities, and to stratification of cancer patients by their survival. The authors benchmark their method against several other widely used methods and show a marked improvement in sensitivity and accuracy. They are also able to show that the differential response of cell lines to drugs can be explained in some cases by the predicted effects of cell line specific mutations. Potentially, this method will find multiple research applications, most importantly the identification of drug response biomarkers for precision medicine.

We thank the reviewer for his kind words and we are glad that he finds our work so interesting and useful.

My enthusiasm for this paper is tempered by difficulties understanding some of the key assertions due to poor writing.

We apologize for this. We have thoroughly revised the manuscript to improve clarity.

There are numerous examples of this. I will pick out a few that stood out:

"These responsive genes are specific to the perturbed pathway (Supplementary Fig. 3) and do almost not overlap with genes that comprise it (Supplementary Fig. 4)" This sentence doesn't parse, and the easiest interpretation, which is that genes that respond to a pathway perturbation are usually not contained in the pathway, is hard to comprehend.

This is now rewritten to:

These responsive genes are specific to the perturbed pathway and have little overlap with genes encoding for its signaling proteins (Supplementary Fig. 4).

Another example:

"Adrenocortical Carcinoma (ACC) shows a significant survival increase with p53 activity ($FDR < 10^{-3}$), supported by the fact that it not harbor any previously reported 6 gain-of-function TP53 variants."

This is now rewritten to:

Adrenocortical Carcinoma (ACC) shows a significant increase of survival with p53 activity ($FDR < 10^{-3}$). This positive effect of p53 on survival is supported by the fact that ACC samples do not harbor any previously reported gain-of-function TP53 variants²⁹

And another:

"For the pan-cancer cohort, we regressed out the effect of the study and age of the patient, and fitted the more for each pathway and method used."

This is now rewritten to:

For the pan-cancer cohort, we fitted the model for each pathway and method separately, regressing out the study of origin and age of the patient.

I would like to see a resubmission of this paper after a careful re-read by a native English speaker and correction of these

We appreciate the reviewer's points about some phrasing errors that happened during our internal revisions. We apologize, they are fixed now.

Methodologically, the main problem that occurred to me during review is that the testing data sets from GDSC and TCGA have previously been submitted to ArrayExpress and GEO and may have contaminated the training set. However, review of the supplementary materials revealed that this is not the case. A statement to this effect would help future readers.

We agree with the reviewer; we also now explicitly state that the three data sets do not overlap.

When comparing PROGENy to other pathway-based activity prediction methods, it would be informative to directly describe the overlap between the genes that define a pathway set in each of the other methods (example, the EGFR Reactome genes) and those that have the strongest coefficients in the PROGENy linear regression. It is likely that the most of the differences in predictive power have to do with the fact that curated pathways focus on post-translational events and are relatively weak with respect to describing downstream transcriptional effects.

We have now expanded our depiction of the overlap between different gene sets in the methods we used to include other signatures to be able to compare the relative overlap (Supplementary Fig. 4).

Reviewer #3 (Remarks to the Author):

The paper touches upon important point, which is the usefulness of gene expression signatures learned from perturbation experiments, and specifically of focusing on the core common to many different contexts. These points are not novel, and very similar approaches have been reported before, but the study is comprehensive, the data available today allows more accurate signatures than before, and the authors apply it to several interesting analyses, showing relevance to drug response and patient prognosis. If the points below are addressed well, I think the paper should be considered for publication.

Major remarks:

1. The authors compare their method, which is based on learning gene signatures from experimental data, to general pathway analysis approaches, which relies only on the knowledge of the pathway and does not require many controlled experiments (nor are limited to signaling pathways which this paper focus on). While they mention that others have tried to learn signatures from perturbations before, they make it sound as if it was limited to breast cancer and proven useless, where in fact these have been widely used successfully in many contexts (such as PMID: 18937865, PMID: 17008526, PMID: 27098033, PMID: 22549044, PMID: 20215513, PMID: 20591134, PMID: 18652687 and many more). This should be better reflected in the text, and in addition the authors should compare their method to these approaches, and not only to methods of applying generic genes sets and pathways.

It is true that many signatures have been published that contrasted two different conditions in terms of their gene expression. We included 208 of such ArrayExpress submissions, many of which have been published as gene expression signatures (we now link to the respective publications in Supplementary Note 2). The potential problem with those differentially expressed genes is that they are heterogeneous up to a point where they provide no clear pattern which pathway was perturbed.

In terms of the previous publications that the reviewer lists, we agree that we should better clarify how our study relates and goes beyond them, which in brief is as follows:

- 18937865: This presents an alternative way to derive consensus signatures; unfortunately, it is not readily available as a tool to use it for comparison. In addition, the nature and use of the signatures is different: PROGENy derives consensus signatures from many cell lines, while this study uses many (~600) perturbations, but on only 3 cell lines. The authors also look at patients, but not in a systematic way, and they do not relate signatures to mutational drivers or drug efficacy, main aspects of our study.
- 17008526: This is the original Connectivity Map; it contains gene expression upon drug treatment, however, the drugs used are (in almost all cases) not specific pathway modulators
- 27098033: This is not a perturbation-response but rather a steady state signature
- 22549044: We already include this experiment in our set (E-GEOD-32975)
- 20215513: This is not a perturbation-response but rather a steady state signature

- 20591134: This experiment does not publish gene expression data; the signatures themselves have been redone later in Gatzka et al. (2009) that we discuss in the manuscript
- 18652687: This experiment does not publish raw gene expression data and hence we can not compare to it or include it. However, we include similar perturbations (a PIK3CA^{H1047R} activation mutant and the inhibitors GDC0941, BKM120, PI-103, among others)

We have revised the introduction and discussion in order to reflect this.

In addition, the reviewer is correct that we have not compared PROGENy to specific published pathway signatures. In order to amend this, we have now extended our functional evaluation (mutations, drug response, and patient survival in Figures 3-6) to also include SPEED-derived Iorio et al. and the Gatzka et al. signatures that we reference.

If they want also to compare to such generic methods, they should first clearly state the difference in approaches, their advantages and disadvantages, and also use these methods to deduce pathway scores of the gene sets they learned from the perturbation data, otherwise it is actually mostly a comparison of the Reactome/KEGG gene set to their gene set.

We included a more detailed description of the advantages and disadvantages of pathway mapping vs. signatures in the introduction. We also perform Gene Ontology enrichment using our signature genes to better characterize the processes involved (now in Supplementary Fig. 5).

We would, however, like to point out that SPIA, Pathifier and PARADIGM are not just gene set methods and require a graph representation of the pathways, so that we do not compare only against other gene sets.

2. The novelty of the research may lie in the claim of a robust method to infer a common core of the pathways they study. As such, they should validate this claim in a more rigorous manner. One thing that is clearly missing is validating the signatures on a validation set not used to learn these signatures.

We agree that the robust method to compute signature of pathways is a main component of our study, although we believe that the curation effort of obtaining a much larger collection of perturbation-induced gene expression changes than previous studies (e.g. SPEED and Gatzka et al.; Supplementary Fig. 2) is also a major contribution towards our functional associations.

Nevertheless, the reviewer is right that we should further validate our signatures. In order to do so, we now (1) provide an additional orthogonal validation using phosphorylation measurements in experiments not included in model building (Fig. 2c and d, Supplementary Fig. 7), (2) performed novel experiments in HEK293 cells where we validated signatures with phosphorylation data (Fig 2b), and (3) perform leave-one-out cross-validation when assessing PROGENy on our curated set of experiments (Fig. 2a and Supplementary Fig. 6a).

A deeper look into the biological meaning of the key genes in their signatures, showing that many of them correspond to known core members of these pathways will also help support this claim.

We thank the reviewer for this comment. We now characterize the 100 genes that comprise each pathway using Gene Ontology enrichment to show the processes that are driven by pathway activation (Supplementary Fig. 5).

More generally all the analyses done on the same data used to derive the signatures (e.g. the results shown in Figure 2) should be done on an independent validation set.

We agree with the reviewer that a completely separate set of perturbation experiments would be ideal to assess model performance. We now validate our model with completely independent gene expression and pathway activity measurements (Fig. 2c and 2d, Supplementary Fig. 7).

Similarly, the Cox proportional hazard model should be learned on a training set and applied on a disjoint validation set.

For TCGA mutations, drug associations and Cox survival models we actually do not learn any model parameters in these data sets. Nevertheless, we now validate our associations for overlapping drugs in the CCLE (Supplementary Table 7) and assess the stability of the survival associations in the TCGA by bootstrapping (Supplementary Table 8). For the survival associations, we were unfortunately not able to use an independent data set as we could not find independent cohorts for the three cancer types we highlight (ICGC largely overlaps with TCGA data; cBioPortal does not list additional cohorts; projects like the AACR GENIE do not have gene expression data to derive our pathway scores from).

Reviewers' comments:

Reviewer #1 (Remarks to the Author):

The authors have replied to my concerns.

Reviewer #2 (Remarks to the Author):

I have reviewed the revised manuscript with respect to the points raised by my own and the other three reviewers. I feel that the authors have done a credible job addressing most if not all of the concerns. In particular, the addition of direct measurements of the predicted phospho-protein changes in response to perturbations of signaling pathways in the HEK293 cell line greatly improves the impact of the paper.

The specific comments I made with respect to language and readability have been addressed.

I am comfortable recommending this paper for publication.

Reviewer #3 (Remarks to the Author):

The manuscript has indeed improved, but there still significant concerns, mostly revolving around how this proposed method is better than previous work, especially studies that are based on a similar approach of gene-expression derived signatures:

1. The authors should state what makes their method better than previous similar approach. While the revised version now include comparison to results of a few previous approaches, it is not clear in what their proposed method is better – is it simply the inclusion of additional data? Is it something about their computational approach? What exactly? Is it the way they learn they signatures or the way they apply them to deduce activity? They should make their claims clear and prove them.

2. The authors should test the different contribution of signature learning and application with other approaches, using their approach to predict activation based on signatures other have learned, and more importantly use other methods to predict activation based on the signatures the authors have derived. Their response to this previous request (point 1 of reviewer #3) is missing the point, I hope this comment is more clear now, and will be taken more seriously. The response states they performed Gene Ontology enrichment using their signature genes to better characterize the processes involved, which is nice but unrelated as they did not use these signatures to predict activation by other means, hence it is not clear if their prediction algorithm (regardless of the gene sets used) is better or worse than other previously suggested methods. They claim Pathifier requires graph representation of the pathways, but this is not the case, and it could be applied on any gene set, and there are also other published methods and approaches they can apply on the gene sets they have learned. They claim they revised the introduction to include a “more detailed description of the advantages and disadvantages of pathway mapping”, but they do not state anywhere that purpose of methods such as Paradigm and Pathifier is to derive activation or deregulation scores of hundreds of pathways, and not just a few select well-studied signaling pathways for which expression based signatures are indeed more suited. As a side note, the point of the few papers mentioned was not that they should or should not include that data in their analysis, but that they should acknowledge previous work that suggested very similar approaches to gene expression derived signatures.

3. Line 68-71. This part is not clear. What do they mean by “those different experiments, however, are heterogeneous”? How is this heterogeneity measured and why does this render previous work non-relevant? Can they show that their method solves this issue and does better than some of the seminal work in the field (e.g. Bild et al. Nature 2006?). It is also not clear what is the point of supplementary fig. 1. Why does inability to cluster by tSNE suggests that their pathway activity

predictions are inconsistent? I'm also a bit confused because a very similar plot in the earlier submission (figure 2a) based on their consensus pathways with similar separation was used to suggest that their consensus pathways are working well.

Response to Reviewers

Reviewer #3 (Remarks to the Author):

The manuscript has indeed improved, but there still significant concerns, mostly revolving around how this proposed method is better than previous work, especially studies that are based on a similar approach of gene-expression derived signatures:

1. The authors should state what makes their method better than previous similar approach. While the revised version now include comparison to results of a few previous approaches, it is not clear in what their proposed method is better – is it simply the inclusion of additional data? Is it something about their computational approach? What exactly? Is it the way they learn they signatures or the way the apply them to deduce activity? They should make their claims clear and prove them.

2. The authors should test the different contribution of signature learning and application with other approaches, using their approach to predict activation based on signatures other have learned, and more importantly use other methods to predict activation based on the signatures the authors have derived. Their response to this previous request (point 1 of reviewer #3) is missing the point, I hope this comment is more clear now, and will be taken more seriously. The response states they performed Gene Ontology enrichment using their signature genes to better characterize the processes involved, which is nice but unrelated as they did not use these signatures to predict activation by other means, hence it is not clear if their prediction algorithm (regardless of the gene sets used) is better or worse than other previously suggested methods.

In her/his points #1 and #2 above, the reviewer raises two valid questions: (1) whether the number of included signatures or the way we derive our scores improves over previous consensus signature methods (SPEED, EPSA) and (2) how, given the genes we found in our signatures, alternative methods of computing pathway scores perform.

To address these points:

(1) We have extended our comparison to include (a) SPEED with their original set of signatures, (b) SPEED with our set of signatures, and (c) EPSA using our set of signatures.

We now show (Supplementary Fig. 6a) that

(a) PROGENy outperforms the original SPEED using their signatures

(b) PROGENy performs similarly well (binomial test; $p=0.5$) to the original SPEED, but SPEED relies on lists of differentially expressed genes, and is hence unsuitable for panels of samples as those in our analyses;

(c) PROGENy performs better than EPSA (binomial test; $p<0.05$) on perturbations and also better than our previous modification of SPEED to make it suitable for panels (Iorio et al.; cf. Fig. 3, 4, and 5).

Furthermore, we quantify the importance of the number of perturbation experiments using PROGENy and find that indeed an increased number of experiments included to derive the signature leads to an improved performance (Supplementary Fig. 6b).

(2) We apologize for misunderstanding the previous comment (point 1 of reviewer #3). We have now also included a comparison to GSEA on our 100 signature genes (using the p-value of the KS-statistic; Supplementary Fig. 6a). We show that GSEA performs similarly overall but is prone to infer the wrong sign of pathway activation (cf. PI3K, VEGF panels and Supplementary Fig. 2b), as our gene sets rely on the sign of the z-scores to distinguish between up- and downregulation (as is SPEED).

To complement this, we already compare to published high quality signatures (Gatza et al.) in Fig. 3, 4 and 5.

They claim Pathifier requires graph representation of the pathways, but this is not the case, and it could be applied on any gene set, and there are also other published methods and approaches they can apply on the gene sets they have learned.

We did not intend to imply Pathifier uses an external pathway structure, but we agree that the statement about Pathifier in the legend of Fig. 1a can be misleading.

We have grouped methods that make use of structure in some capacity (Pathifier infers signal flow, it does not merely use the set as a set with equal weights) to:

Reasoning about pathway activation. Most pathway approaches make use of either the set (top panel) or infer or incorporate structure (middle panel) of signaling molecules to make statements about a possible activation [...].

They claim they revised the introduction to include a “more detailed description of the advantages and disadvantages of pathway mapping”, but they do not state anywhere that purpose of methods such as Paradigm and Pathifier is to derive activation or deregulation scores of hundreds of pathways, and not just a few select well-studied signaling pathways for which expression based signatures are indeed more suited.

This is a good point. We have now extended the paragraph in the introduction to not only refer to previous work, but also state that the primary limitation of perturbation-based methods is the number of signatures for which there is enough experimental data allowing to build consensus models.

In the results, we further refine this statement to compare performance of the actual method to EPSA and SPEED (which are the two methods for consensus signatures, i.e. similar approaches to ours):

PROGENy separates basal and perturbed arrays better (Supplementary Table 3; binomial test; $p < 0.04$) than EPSA on our curated set of experiments and in addition to SPEED also infers the sign of pathway activity (Supplementary Fig. 6).

In order to address the fact that expression-based methods can be more easily applied to any number of pathways, we have added the following statement in the introduction:

While these methods can be applied to almost any pathway, they are based on mapping transcript expression to the corresponding signaling proteins and hence do not take into account the effect of post-translational modifications (Fig. 1a).

And in the discussion:

The latter can be used for many more pathways, as information on the pathway components is more often available than perturbation experiments. However, our results indicate that one should be cautious when interpreting the expression level of a pathway as its activity.

As a side note, the point of the few papers mentioned was not that they should or should not include that data in their analysis, but that they should acknowledge previous work that suggested very similar approaches to gene expression derived signatures.

We thank the reviewer for the clarification. Besides the individual studies we discuss (Connectivity Map, Bild, Gatzka, EPSA, SPEED), we now cite a general review on this topic.

3. Line 68-71. This part is not clear. What do they mean by “those different experiments, however, are heterogeneous”? How is this heterogeneity measured and why does this render previous work non-relevant?

For clarity, we have rewritten the paragraph to:

However, the same signaling pathways may trigger different downstream gene expression programs depending on the cell type or the perturbing agent used. Hence, if gene expression signatures are to be used as a generally applicable pathway method, there is a need to address this context specificity.

Can they show that their method solves this issue and does better than some of the seminal work in the field (e.g. Bild et al. Nature 2006?).

We agree with the reviewer that we should do additional comparisons, in particular: (1) compare the consensus signature of our method to previously published consensus methods in terms of how well they represent pathway activation in different conditions and (2) compare the GDSC and TCGA application of our consensus to previously published individual signatures.

For point 1, we now compare to EPSA and SPEED using our curated set of experiments (cf. response above).

For point 2, we use the high quality signatures of Gatzka *et al.* instead of Bild *et al.* because they used the same experiments and the latter published neither their microarray data nor an implementation of their method.

It is also not clear what is the point of supplementary fig. 1. Why does inability to cluster by tSNE suggest that their pathway activity predictions are inconsistent? I'm also a bit confused because a very similar plot in the earlier submission (figure 2a) based on their consensus pathways with similar separation was used to suggest that their consensus pathways are working well.

We intended to show a better pathway clustering by PROGENy pathway scores than differential expression of genes.

While we did not intend to claim that pathway activity predictions are inconsistent but rather the global gene expression differences in the perturbation experiments are, we agree that this plot did not convey the message sufficiently well.

As Supplementary Fig. 5 already shows the relative activation patterns of the inferred pathway scores for different methods, we removed Supplementary Fig. 1.

REVIEWERS' COMMENTS:

Reviewer #3 (Remarks to the Author):

The authors have replied to my concerns.